# Lactylation: A Novel Post-Translational Modification with Clinical Implications in CNS Diseases

**DOI:** 10.3390/biom14091175

**Published:** 2024-09-19

**Authors:** Junyan Liu, Fengyan Zhao, Yi Qu

**Affiliations:** 1Department of Pediatrics/Key Laboratory of Birth Defects and Related Diseases of Women and Children (Ministry of Education)/NHC Key Laboratory of Chronobiology, West China Second University Hospital, Sichuan University, Chengdu 610041, China; liujunyan1986@alu.scu.edu.cn (J.L.); zhaofengyan@scu.edu.cn (F.Z.); 2Neonatal Intensive Care Unit, Binzhou Medical University Hospital, Binzhou 256600, China

**Keywords:** lactylation, epigenetics, neurodevelopment, central nervous system diseases, therapeutic target

## Abstract

Lactate, an important metabolic product, provides energy to neural cells during energy depletion or high demand and acts as a signaling molecule in the central nervous system. Recent studies revealed that lactate-mediated protein lactylation regulates gene transcription and influences cell fate, metabolic processes, inflammation, and immune responses. This review comprehensively examines the regulatory roles and mechanisms of lactylation in neurodevelopment, neuropsychiatric disorders, brain tumors, and cerebrovascular diseases. This analysis indicates that lactylation has multifaceted effects on central nervous system function and pathology, particularly in hypoxia-induced brain damage. Highlighting its potential as a novel therapeutic target, lactylation may play a significant role in treating neurological diseases. By summarizing current findings, this review aims to provide insights and guide future research and clinical strategies for central nervous system disorders.

## 1. Introduction

Traditionally, the glycolytic metabolism of glucose to lactate has been associated with anaerobic or hypoxic conditions. In the 1920s, Otto Warburg discovered that rapidly proliferating tumor cells prefer glucose uptake and fermentation even in the presence of oxygen, leading to the intracellular and extracellular accumulation of lactate [1]. This phenomenon, known as aerobic glycolysis, results from metabolic reprogramming that suppresses mitochondrial oxidative phosphorylation (OXPHOS) [2]. Lactate plays dual roles as an energy substrate and a signaling molecule within a shuttle system between different cell populations. Various cell types, including neurons, cardiomyocytes, and tumor cells, can uptake lactate, convert it to pyruvate under aerobic conditions, and integrate it into the tricarboxylic acid (TCA) cycle for complete oxidation, yielding ATP. Moreover, lactate functions as a signaling molecule and modulates processes such as cell proliferation, metabolism, angiogenesis, invasion, and immune responses [3,4,5,6].

In the central nervous system (CNS), lactate regulates neuronal excitability, synaptic plasticity, and learning and memory processes through specific transporters and receptors [7]. In 2019, Zhang et al. [8] discovered that lactate induces a novel post-translational modification called lactylation, significantly impacting gene regulation and cellular functions. Elevated lactate levels and histone lactylation were observed in an M1 polarization model of macrophages stimulated by bacterial infection. Histone lactylation is particularly enriched at the promoters of M2-like homeostatic genes (e.g., *Arginase-1* and *Vegfa*), thereby modulating transcription and promoting tissue repair and wound healing. Emerging evidence indicates that metabolic reprogramming leading to lactate production and subsequent lactylation occurs in various cells and diseases, including pulmonary hypertension, heart failure, renal fibrosis, atherosclerosis, neurodegenerative diseases, psychiatric disorders, trauma, and hypoxic/ischemic injuries. Lactylation is involved in numerous physiological and pathological processes, such as cell fate determination, neuronal excitability, oxidative stress response, and immune regulation [6,9]. Consequently, research on lactylation has gained momentum, particularly in the CNS, where it influences neural development, neuropsychiatric disorders, glioblastomas, and hypoxia-related brain injuries [10,11,12,13].

This review systematically summarizes the regulatory roles and mechanisms of lactation in CNS development and diseases, offering valuable insights and targets for future research and clinical treatment.

## 2. Lactate Metabolism and Shuttling in the Brain

Since lactate was first reported in 1780, scientific research has significantly advanced our understanding of its role in metabolism [14]. Under normoxic conditions, cells metabolize glucose to pyruvate, which enters the TCA cycle to produce water, carbon dioxide, and ATP, meeting cellular energy demands. During hypoxia, glycolysis is activated, and pyruvate is reduced to lactate by lactate dehydrogenase (LDH) to generate ATP. Initially considered a metabolic waste product, lactate is now recognized as a vital component cleared primarily by the liver through gluconeogenesis (Cori cycle) and OXPHOS (Krebs cycle) [15]. Notably, many tumor cells rely on aerobic glycolysis for energy production, fermenting glucose to lactate even in the presence of oxygen—a phenomenon known as the Warburg effect [16]. Lactate, as a product of glycolysis and a substrate for mitochondrial respiration, bridges the glycolytic and aerobic pathways. The lactate shuttle hypothesis posits that lactate transcends compartmental barriers, shuttling within and between cells, tissues, and organs, playing critical roles in energy metabolism, immune responses, memory formation, wound healing, and tumorigenesis [3,6].

### 2.1. Lactate as an Energy Substrate in the Brain

Neurons, which are characterized by high metabolic activity, exhibit significant energy demands. Although the brain constitutes only 2% of body weight, it accounts for approximately 20% of the body’s oxygen and energy consumption, primarily for maintaining membrane potentials, synaptic transmission [17], and neurotransmitter synthesis and release. Brain activation leads to dynamic changes in energy demand, necessitating those neurons meet both sustained and transient high-energy demands [18]. Under resting conditions (normoxia), neurons primarily derive energy from aerobic glucose oxidation, whereas astrocytes rely on aerobic glycolysis. Metabolic remodeling during neuronal activation increases glycolytic activity [19,20]. Activated neurons release glutamate at synapses, which is then taken up by astrocytes, promoting glucose uptake and lactate production. Neurons then utilize lactate from astrocytes as an energy substrate through the astrocyte-neuron lactate shuttle (ANLS) [21] (Figure 1).

Isotopic labeling of lactate and glucose has shown that lactate enters neurons and is converted to pyruvate by highly expressed LDH1, which is a key step in lactate as an energy molecule [22,23]. During high neuronal activity, lactate serves as a faster and more direct energy source than glucose [24]. Studies on cultured neurons, brain slices, and in vivo models have revealed that lactate sustains neuronal vitality and function during activation more effectively than glucose. Higher lactate levels were observed in astrocytes than in neurons in the mouse brain, and a higher cytosolic NADH/NAD+ ratio was observed in astrocytes in hippocampal slices, supporting greater lactate production in astrocytes and indicating net lactate transport from astrocytes to neurons [19]. The specific expression of glycolytic enzymes in astrocytes, such as fructose-2,6-bisphosphatase 3 (PFKFB3), enhances the activity of the glycolysis-limiting enzyme phosphofructokinase (PFK), enabling high glycolysis levels in astrocytes [25], making astrocytes a primary lactate synthesizer converting glucose from the bloodstream or stored glycogen into pyruvate and lactate [26,27].

The relative contributions of glucose and lactate as primary energy substrates, along with their respective metabolic pathways (glycolysis and OXPHOS in neurons), remain subjects of ongoing debate [18,28]. Research has shown that substituting or supplementing glucose with pyruvate or lactate in artificial cerebrospinal fluid significantly reduces NADH overshoot and oxygen consumption during neuronal network activation. Both neurons and astrocytes consume energy during this period, with neurons utilizing glucose as a crucial energy source [29]. These findings challenge the view that astrocyte-derived lactate is the primary energy substrate for neurons but do not directly elucidate the relative contributions of glycolysis and other metabolic pathways to energy provision. Transient changes in the NADH/NAD+ ratio of individual neurons after stimulation were measured in the brains of awake mice using genetically encoded fluorescent biosensors. These measurements revealed that the transient changes were caused by a rapid increase in cytosolic glycolysis. Blocking lactate entry into neurons via monocarboxylate transporters (MCTs) during the resting state decreases neuronal NADH levels. However, during neuronal activation, the rapid increase in the NADH/NAD+ ratio is not inhibited by MCT blockers, supporting the presence of ANLS in the resting state but not during activation [17].

Based on a comprehensive analysis of past and recent studies, lactate has both potential and limitations as a supplementary fuel for synaptic transmission and neuronal network oscillation. During low-energy-demand network activities, such as in anesthetized neurons, lactate can effectively substitute for glucose and may even demonstrate higher efficiency. However, in high-energy-demand network activities, lactate can only serve as a supplementary fuel, partially replacing glucose [30]. This indicates that the brain’s energy sources are diverse, including glucose and lactate, and that metabolic pathways are not singular; the utilization of specific energy substrates depends on the brain’s ongoing demands and substrate availability. In different pathophysiological contexts, neurons optimize their overall network function by utilizing the best metabolic pathways and energy substrate combinations based on their activity status. These potential mechanisms warrant further investigation.

### 2.2. Lactate as a Signaling Molecule in the Brain

In the ANLS paradigm, lactate functions not only as a neuronal energy substrate but also as a signaling molecule that regulates neural excitability, synaptic plasticity, and memory consolidation [31,32]. The lactate shuttle flux is modulated by lactate concentration gradients, pH gradients, and redox states. Lactate transport between neurons is facilitated by MCTs and gap junction hemichannels. MCT1, MCT2, and MCT4 are extensively expressed in the brain. MCT1 is predominantly localized in vascular endothelial cells, astrocytes, and oligodendrocytes; MCT2 is primarily found in neurons, particularly in dendritic spines; and MCT4 is almost exclusively expressed in astrocytes [33]. The high affinity of MCT2 for lactate ensures efficient lactate uptake by neurons [34]. CD147 functions as an essential chaperone molecule for MCT1 and MCT4, stabilizing their structure and facilitating translocation and localization to the cell membrane [35]. Gap junction hemichannels, composed of connexins such as connexin 30 and 43, enable extensive astrocyte communication and network synchronization [36,37]. The G protein-coupled receptor GPR81, also known as hydroxycarboxylic acid receptor 1(HACR1), acts as an endogenous lactate receptor and is expressed in neurons of the cerebral cortex and hippocampus at excitatory post-synaptic membranes and is enriched in the blood–brain barrier [38]. Together, MCTs, gap junction hemichannels, and GPR81/HCAR1 form the molecular foundation of lactate signaling (Figure 1).

Recent research highlights the role of lactate in neuronal excitability. Lactate dehydrogenase A (LDHA) in glial cells of the dorsomedial prefrontal cortex (mPFC) is instrumental in regulating lactate homeostasis, thereby influencing neuronal excitability and depression-like behavior [23]. Experimental evidence indicates that LDHA knockdown reduces lactate production, leading to decreased neuronal excitability and exacerbated depression-like behavior in mice subjected to social defeat. Conversely, LDHA overexpression enhances lactate production, increases neuronal excitability, and mitigates depression-like phenotypes. The administration of lactate in mouse models of depression has therapeutic effects on depressive behaviors, underscoring the pivotal role of astrocyte-derived lactate in modulating neuronal excitability. Further studies have demonstrated that restricted lactate shuttling diminishes neuronal excitability. Specifically, the inhibition of MCT2 expression in neurons prevents restoration of neuronal firing or alleviation of depressive-like phenotypes despite exogenous lactate supplementation. Mechanistic studies have revealed that lactate enhances neuronal excitability by inhibiting rapid after-hyperpolarization, an effect mediated by large-conductance Ca^2+^-activated potassium (BK) channels. Additional research has indicated that blocking gap junction hemichannel proteins significantly reduces synaptic excitability in hippocampal neurons. The astrocyte-specific knockout of connexin 43 impairs lactate shuttling, decreasing the excitability of arousal-promoting orexin neurons and causing excessive sleepiness and fragmented arousal in mice [36]. These findings suggest that lactate inhibits function in preventing neuronal hyperactivation, while this claim requires further evidence to be substantiated. Moreover, lactate modulates neuronal excitability via HCAR1 activation, reducing presynaptic spontaneous calcium-spiking activity. In epilepsy models, HCAR1 knockout mice exhibit reduced seizure thresholds, increased severity, and prolonged duration, similar to LDHA inhibition [39,40]. HCAR1 may regulate neural activity through functional interactions with other Gi-coupled receptors via the Giα and Giβγ pathways [41]. Further experiments are needed to confirm this. Lactate has also been shown to induce glutamatergic synaptic enhancement in hippocampal CA3 pyramidal cells, an effect that is synapse-specific and dependent on N-methyl-d-aspartic acid receptor (NMDARs) activation, cholera toxin-sensitive G-protein-coupled receptors, and post-synaptic calcium accumulation [42]. In conclusion, lactate plays a critical role in regulating neuronal excitability and synaptic activity, though further studies are needed to clarify the precise mechanisms underlying these effects (Figure 1).

Synaptic plasticity represents a dynamic alteration in the strength and architecture of synaptic connections that underpin learning and memory processes. During early development, the human brain exhibits high levels of aerobic glycolysis and substantial synaptic proliferation, peaking around five years of age. In adulthood, aerobic glycolysis remains active only in the regions expressing juvenile genes, with a decline in synaptic plasticity-related gene expression, linking glycolysis to synaptic plasticity [43]. Inhibition of astrocyte glycogen phosphorylase using 1,4-dideoxy-1,4-imino-D-arabinitol (DAB) impedes lactate production and suppresses long-term potentiation (LTP) and memory consolidation during training in the rat hippocampus. Conversely, exogenous lactate restores these processes, whereas glucose does not, suggesting that the role of lactate extends beyond merely serving as an energy substrate [44]. In the step-through inhibitory avoidance (IA) paradigm, training mice showed upregulated expression of genes associated with astrocyte-neuron metabolism, notably MCT1 and MCT4. Mice deficient in MCT1 and with suppressed MCT2 expression exhibit compromised memory [45,46]. Another study revealed that diminished expression of MCT4 in dorsal hippocampal astrocytes and MCT2 in neurons impairs spatial learning and memory reversible by lactate administration if MCT2 expression remains intact [47]. MCT-mediated lactate transport in the hippocampus is crucial for learning spatial tasks. Inhibition of MCT1, MCT2, and MCT4 expression, along with DAB injection, attenuates methamphetamine (METH)-induced conditioned place preference (CPP), plasticity-related gene expression, neuronal Ca^2+^ levels, and synaptic structure. Exogenous lactate reverses these effects except when MCT2 is inhibited [48]. These findings indicate that astrocyte-derived lactate, transported to neurons via MCT2, regulates synaptic activity and memory consolidation [7]. Molecular mechanisms reveal that lactate enhances synaptic remodeling and memory consolidation by increasing synaptic remodeling gene expression [31,44,49]. The expression of pCREB (phosphorylated cAMP response element binding protein) and Arc (activity-regulated cytoskeletal protein), along with phosphorylated cofilin, represents key mechanisms underlying long-term synaptic plasticity and memory formation, as well as the associated synaptic structural changes, all of which hinge on lactate transport between astrocytes and neurons [44]. NMDAR activation by glutamate and glycine synergistically increases intracellular calcium concentration, Erk1/2 phosphorylation, and immediate early gene (IEG) expression such as *Arc* and *Zif268* [50]. Lactate metabolism elevates the intracellular NADH/NAD ratio, enhancing neuronal redox state and potentiating NMDAR activation and subsequent Erk1/2 signaling through redox-sensitive NR1 subunits, promoting IEG expression. Specific inhibitors MK801 and U0126 abolish lactate-induced IEG expression [49]. cAMP, a pivotal intracellular second messenger, integrates signals from multiple G protein-coupled receptors (GPCRs) [51]. Lactate may facilitate synaptic remodeling and memory consolidation through GPCR activation and downstream cAMP/protein kinase A (PKA) signaling pathways; however, further experiments are needed to validate this. Activation of cAMP leads to the targeting of PKA to key proteins involved in hippocampal synaptic plasticity and memory storage [52,53]. In summary, lactate plays an essential role in synaptic remodeling and memory consolidation via multiple signaling pathways, including IEG expression promotion through NMDARs and Erk1/2 signaling cascades, as well as the GPCR/cAMP/PKA pathway (Figure 1).

## 3. Protein Lactylation in the Brain

### 3.1. Mechanisms of Protein Lactylation

Intermediate metabolites and end products of cellular metabolism, such as acetyl-CoA, succinyl-CoA, and acyl-CoA, play crucial roles not only in metabolic processes but also in regulating cellular activities [54]. In chromatin epigenetic regulation, these metabolites can covalently modify protein residues, influencing epigenetic states [55,56]. Lysine acetylation, a common post-translational modification (PTM), exemplifies this regulatory mechanism by altering protein polarity and function through modification of the ε-amino group of lysine [56]. Lactate, a chiral molecule, exists in three optical isomers: d-lactate, l-lactate, and racemic dl-lactate, with L-lactate (henceforth referred to as lactate) being the predominant form [57]. Recent research has identified lactate as a multifunctional molecule involved in metabolic reprogramming and epigenetic modifications, thereby significantly influencing cellular function.

Building on the concept of lysine acetylation, Yingming Zhao’s team hypothesized in 2019 that lactate could act as a substrate for PTMs, modifying lysine residues on histones to regulate gene expression [8]. They were the first to identify lysine lactylation (Kla) as a novel histone modifier. Initial evidence of histone lactylation was obtained from human MCF-7 cells, where a 72.021 Da mass shift in core histone lysine residues digested by trypsin indicated the addition of a lactyl group to the ε-amino group of lysine residues (Figure 2). This was confirmed using high-performance liquid chromatography (HPLC) and tandem mass spectrometry (MS/MS) to compare the elution times and fragmentation patterns of the synthetic and endogenous lactylated peptides. Immunoblotting with pan-anti-lactylation antibodies and metabolic experiments using synthetic stable isotope-labeled lactate further validated histone lactylation (Figure 3). Their findings revealed a direct correlation between histone lactylation levels and local lactate concentrations, particularly under stress conditions such as hypoxia and bacterial infection. These results highlight the nuanced role of lactate in cellular regulation [8].

The research team led by Zhao et al. proposed that lactyl-CoA, an activated form of acetic acid, is transferred to lysine residues, serving as a substrate for lactylation. This enzymatic reaction directly stimulates gene transcription in the chromatin. In early 2020, another research team published further evidence of protein lactylation and its involvement in epigenetic regulation [58]. They suggested that methylglyoxal, a byproduct of high glycolytic activity, binds to glyoxalase 1 (GLO1) to form glyoxalase glutathione (LGSH), which serves as a substrate for lactylation. The lactyl group is then transferred to protein lysine residues via non-enzymatic acyl transfer, resulting in lactylation (Figure 2). Glyoxalase 2 (GLO2) hydrolyzes LGSH to glutathione and d-lactic acid. Knockout experiments of GLO2 demonstrated significant increases in LGSH and lactylation levels in cells.

Research indicates that lactylation predominantly enriches glycolytic enzymes such as GLO1 and non-histone proteins with glyoxalase domains, including glyoxalase 4 (GLOD4). Gaffney et al. [58] proposed that protein lactylation is a passive, non-enzymatic reaction occurring on core histones and non-histone proteins within cellular organelles and membranes (Figure 2). Subsequent studies have identified multiple lactylation sites on histones and non-histones across a wide range of organisms, including human tissues, animal models, plants, bacteria, fungi, and parasites [13,59,60,61,62,63], underscoring the pervasive nature of lactylation. In summary, lactylation is an emerging post-translational modification that adds a new layer of complexity to epigenetic regulation.

### 3.2. Regulation of Protein Lactylation

PTMs enhance protein functionality and can be regulated either enzymatically or non-enzymatically, with enzymatic regulation being the most prevalent. Under enzymatic conditions, protein acetylation is controlled by specific acetyltransferases and deacetylases, often referred to as “writers” and “erasers”, which respectively add or remove lactyl groups from amino acid residues such as lysine and glycine. Specific protein domains, known as “readers”, interpret the key epigenetic information conveyed by these modifications [64] (Figure 2).

The unique deep acyl pocket within the p300 active site allows it to bind various acyl groups, positioning p300 as a pivotal writer of numerous protein acylations [65]. Zhang et al. demonstrated that histone lactylation levels in HEK293T cells increase or decrease with the overexpression or knockdown of p300, providing the first evidence that p300 acts as a histone lactylation writer [8]. In addition to histones, multiple studies have confirmed the activity of p300 and its homologous protein, CREB-binding protein (CBP), as non-histone lactylation writers. For instance, in a mouse model of heart failure, p300 overexpression upregulated the lactylation level of α-myosin heavy chain (α-MHC) K1897, alleviating heart failure [66]. Following myocardial infarction, CBP/p300 promotes lactylation of endothelial cell Snail1, contributing to myocardial fibrosis and heart dysfunction [67]. In recent years, evidence has supported the role of p300 in brain cells, particularly microglia. Lin et al. [68] found that CBP/p300 and its associated factor, PCAF, are writers of histone lactylation in mouse microglial cell line BV2 and human microglial cell line HMC3, which are involved in aging microglia and Alzheimer’s disease (AD). In a mouse model of glioblastoma (GBM), the p300 inhibitor CPI-1612 was shown to eliminate histone lactylation and its mediated IL10 expression in monocyte-derived macrophages, thereby enhancing the efficacy of immunotherapy for GBM, while acetylation (Kac) modification levels remained unaffected [69]. The gene encoding E1A-binding protein p300 (Ep300) produces a cellular p300 transcript associated with adenovirus E1A. In the TMZ-resistant GBM cell line TBD0220, the binding level of EP300 to the *LUC7L2* promoter was significantly increased, highlighting its role as a histone lactylation writer [70]. During neural development, dynamic changes occur in histone lactylation, Kac, and crotonylation (Kcr) [71]. Considering p300’s role in various PTMs and its involvement in regulating neuronal identity through histone Kac in the adult brain [72], it is hypothesized that p300 also regulates histone lactylation in the developing brain. These findings indirectly confirm p300’s role as a writer in neural cells (Figure 2).

Gaffney et al. [58] demonstrated that GLO2 reduces protein lactylation by preventing the accumulation of LGSH, although the regulatory mechanisms governing the site specificity and abundance of lactylation modifications remain unclear. Jennings et al. [73] employed chemical biology and CRISPR-Cas9 screening to identify NAD+-dependent deacetylase SIRT2 as a potent remover of non-enzymatic acyl groups, marking it as the first recognized “eraser” for non-enzymatic lactoylLys PTMs. This discovery offers deeper insights into the regulation of non-enzymatic PTMs. In neuroglial cells, Han et al. [74] found that SIRT1, SIRT2, SIRT3, and SIRT5 could catalyze the removal of acyl groups using a fluorescent probe with lactyllysine residues, and that SIRT2 demonstrated the highest catalytic activity, which could be inhibited by Tenovin-6. Carlos et al. [75] systematically predicted that Zn^2+^-and NAD^+^-dependent class I histone deacetylases (HDACs) and SIRT1-3 function as deacetylases for lactylation. They confirmed the delactylation activity of HDAC1 and HDAC3 in HeLa cells and demonstrated site-specific erasure effects. The “erasure” of multiple PTMs by HDAC1-3 is critical for neuronal development and differentiation [76]. Dai et al. [71] discovered that knocking down HDAC1-3 or applying their inhibitors significantly increased the expression levels of H3K18la, confirming the de-lactylation role of HDAC1-3 in brain histone lactylation. Their findings also revealed that genome-wide alterations in H3K9ac, H3K9cr, and H3K18la at different developmental stages favor neuronal differentiation, albeit with distinct patterns of change. Considering the pivotal role of CBP/p300 in establishing and maintaining neuronal identity [72], it is likely that the levels of numerous histone PTMs depend on a balance between the “writing” activities of CBP/p300 and the “erasing” activities of HDACs, collectively regulating gene expression to drive neurogenesis and differentiation (Figure 2).

### 3.3. Lactylation and Neural Development

Epigenetic regulatory factors play critical roles in the proliferation and differentiation of neural cells to ensure normal neocortical development. Studies have shown that the chromatin of neural stem cells is condensed and exhibits reduced dynamics, suggesting that these epigenetic factors regulate the expression of specific genes [77]. Complementing this, Dai et al. [71] explored the dynamic changes and functions of histone modifications, specifically Kcr and Kla, during neural development. Their findings revealed that these modifications are prevalent in the embryonic cortex, particularly within the cortical plate region, where H3K9cr and H3K18la levels are significantly elevated compared to those in the ventricular/subventricular zone. These histone marks vary across developmental stages, highlighting the distinct dynamics of site-specific histone modifications during cell fate transition. Notably, the overall levels of Kcr and Kla modifications were not affected by transcriptional inhibition. Their study demonstrated that increased H3K18la enrichment activated genes associated with neuronal differentiation and maturation, whereas decreased H3K18la enrichment downregulated genes related to cell proliferation. Despite these insights, the direct regulatory relationship between H3K18la and neural stem cell fate remains unclear. Furthermore, the mechanisms through which lactylation regulates neural stem cell fate, including potential cooperation with other epigenetic modifications or regulatory factors, and its dynamic regulation over time and space, remain unclear.

During development, cells undergo metabolic reprogramming to perform specific functions at various stages. The role of lactylation in the “epigenome-metabolome-epigenome” crosstalk was elucidated by Liu et al., who proposed a unique three-stage pathway regulated by the maternally expressed transcription factor Glis1 in pluripotent stem cell fate determination [78]. Stage 1 involves Glis1-mediated epigenetic regulation, where Glis1 binds to glycolytic gene promoters and initiates their expression. Stage 2 encompasses metabolic regulation, marked by the upregulation of the glycolysis products acetyl-CoA and lactate. Stage 3 features epigenetic regulation, where acetyl-CoA and lactate mediate histone Kac and lactylation, regulating the expression of pluripotency genes and “second wave” genes. This study underscores the central roles of histone Kac and lactylation in epigenetic regulation driven by glycolytic metabolism.

Cell metabolism exhibits temporal and spatial heterogeneity during embryonic development and plays a crucial role in cellular identity and behavioral transitions. Neural crest cells (NCCs), which are transitional pluripotent stem cells during embryonic development, undergo metabolic reprogramming based on the developmental stage. High glycolytic flux is essential for NCC migration, with enhanced glycolysis and lactate production driving gene expression changes that shape NCC identity and function [79]. In 2024, Prof. Marcos Simoes-Costa’s team at Harvard Medical School constructed and compared whole-genome lactylation maps of NCCs and paraxial mesoderm cells (PSM), another highly glycolytic cell group, alongside single-cell ATAC-seq datasets [11]. Their study revealed widespread lactylation in NCCs, characterized by specific temporal and spatial patterns. Lactylation was prominent in cranial NCCs during the migration (HH9) and premigratory (HH12-13) stages. This spatiotemporal specificity highlights the crucial role of lactylation in NCC development. Lactylation predominantly targets gene loci within the NCC gene regulatory network (GRN), with tissue-specific enrichment of H3K18la on NCC enhancers, marking genes that contribute to NCC-specific features during early development. Specific lactylation gene loci in NCCs exhibit unique accessibility and are dynamically modulated throughout development, with restricted lactylation impairing their accessibility. Moreover, lactylation gene sets in NCCs and PSM were enriched in cell adhesion-related terms, highlighting the importance of the glycolysis-lactylation axis in epithelial-mesenchymal transition (EMT) and cell migration. Lactylation promotes NCC migration by regulating genes that support cell type-specific functions and behaviors. LDHA/B knockout, resulting in decreased lactylation levels, inhibited NCC migration, whereas lactate addition enhanced NCC migration, indicating that lactate generation and glycolytic flux are essential for the activation of NCC GRN. Genomic regions containing *SOX* and *TEAD* motifs tended to be lactylated in NCCs, suggesting a role for SOX9 and YAP/TEAD in lactylation. During embryonic development, the transcription factor Bach1 can suppress the expression of hexokinase 2 (HK2) and glyceraldehyde-3-phosphate dehydrogenase (GAPDH), thereby downregulating glycolysis and lactate production in oligodendrocyte precursor cells. This leads to decreased histone lactylation and *LRRC15* transcription, thereby affecting astrocyte formation. In mice with conditional Bach1 knockout, abnormal neuronal differentiation and anxiety-like behaviors have been observed [80]. These findings revealed the role of specific transcription factors in lactylation regulation, further elucidating how lactylation influences gene expression. Understanding the regulatory role and mechanisms of lactylation in neurodevelopment is complex and crucial, as it offers new insights and approaches for treating neurodevelopmental disorders (Table 1).

### 3.4. Lactylation and Neuropsychiatric Disorders

Neurological and psychiatric disorders are frequently associated with abnormalities in complex metabolic pathways, including mitochondrial dysfunction, glucose metabolism disorders, and defects in neuronal glucose uptake. Elevated brain lactate levels have been observed in various conditions [81,82,83,84], such as severe depression, anxiety, AD, Parkinson’s disease (PD), schizophrenia (SCZ), post-traumatic stress disorder (PTSD), and epilepsy (EP). Research has indicated that histone lactylation is dependent on lactate concentration [8]. Hagihara et al. [81] demonstrated through in vivo and in vitro experiments that lactylation occurs in brain cells, is dose dependent, and is regulated by neural excitation and social stress. Increased lactate, high potassium, or electroconvulsive stimulation (ECS) of neuronal excitability can enhance lactylation levels in mouse brain cells. This correlates with increased expression of the neural activity marker c-Fos and altered social behaviors, including increased anxiety-like behaviors in stress models. This suggests that lactylation in local brain tissue may influence the function of neurofunctional areas. However, further investigations are required to determine whether lactate is a direct source of protein lactylation and to elucidate how protein lactylation-mediated molecular signal transduction is involved in neural activity under physiological and pathological conditions.

AD is the most prevalent neurodegenerative disease, characterized by amyloid-β (Aβ) plaques and tau neurofibrillary tangles, leading to neuronal loss and cognitive decline [85]. Studies have found reactive microglial cells co-located with Aβ plaques in AD, with pro-inflammatory activation preceding plaque formation, indicating early microglial cell neuroinflammation in AD [86,87]. In AD, microglial cells exhibit high glucose uptake and metabolic activity [88], and inflammation induces a metabolic shift from oxidative phosphorylation to aerobic glycolysis [89]. This shift supports immune function but may lead to dysfunction and pro-inflammatory cytokine release. Inhibiting glycolysis can ameliorate inflammation-related diseases [90,91]. Both 5XFAD mice and AD patients showed increased lactate levels. Notably, significant upregulation of H4K12la in microglial cells activates the transcription of glycolytic genes, forming a glycolysis/H4K12la/PKM2 positive feedback loop and exacerbating dysregulation of glucose metabolism and microglial dysfunction [92]. Cell senescence, which is closely related to AD, involves metabolic shifts towards aerobic glycolysis in senescent microglial cells [68,93,94]. Lactate accumulation increases histone lactylation, with H3K18la being significantly upregulated in the hippocampal tissues of naturally aged and AD mice. Enhanced H3K18la binds to the *Rela (p65)* and *NF-κB1 (p50)* promoter regions, activating the NF-κB signaling pathway and upregulating senescence-associated secretory phenotype (SASP) components IL-6 and IL-8, suggesting an H3K18la/NF-κB/SASP positive feedback loop exacerbating brain aging and AD pathology [68]. In the aforementioned AD studies, H4K12la and H3K18la levels were elevated in the hippocampal and cortical tissues of mice, respectively. In AD pathology, H4K12la promotes pathology by regulating key glycolytic enzyme genes (such as *PKM2*), whereas H3K18la acts through the regulation of inflammation-related signaling pathways (such as NF-κB). Targeting these pathways or the metabolic reprogramming of senescent microglial cells presents new insights into the development of novel AD interventions and treatments [95]. Exercise-induced elevation of brain lactate levels can guide microglial cells towards a repair/anti-inflammatory phenotype via histone H3 lactylation, reducing neuroinflammation, improving cognitive function in AD-like mice, and reversing neuronal loss, acting as an “accelerator” of the “lactate clock” in microglia. In vitro experiments confirmed that exogenous sodium lactate treatment of Aβ1-42 or LPS-stimulated BV2 cells increased the expression of repair genes *Arg1* and *VEGF*, indicating that sodium lactate accelerates the transition of BV2 cells towards an anti-inflammatory/repair phenotype [96]. Thus, guiding microglial cells towards a repair phenotype may modulate the progression of AD.

Recent studies confirmed the role of lactylation in other neuropsychiatric diseases. In a PD mouse model, increased lactate production from enhanced glycolysis elevated H3K9la levels, enhanced the transcription of the solute carrier family 7 member 11 (*Slc7a11*) gene, and promoted pro-inflammatory microglial activation. Inhibiting glycolysis with 2-deoxyglucose (2-DG) reduced microglial activation and dopaminergic neuron damage and improved motor function [97]. In an SCZ mouse model, H3K9la and H3K18la were enhanced, and 2-DG reduced lactate accumulation and H3K9la, alleviating behavioral symptoms [98]. In PTSD, lactate activates the ACCN2 channel, found in the amygdala and substantia nigra, worsening symptoms and contributing to pathology by causing HIF-1α lactylation and dysfunction [99]. Patients with epilepsy exhibit reduced MCT1 and MCT2 expression in the microvasculature, which leads to elevated lactate levels. 2-DG partially restores the hippocampal-amygdala circuit relationship induced by epilepsy and mitigates anxiety-like behavior, making lactate receptor signaling a promising target for the development of antiepileptic drugs [100] (Table 2).

### 3.5. Lactylation and Glioblastoma

The Warburg effect leads to lactate accumulation through aerobic glycolysis, a hallmark of metabolic reprogramming in tumor cells. GBM is a fatal primary brain glioma that responds poorly to existing treatments (surgery, chemotherapy, radiotherapy, immunotherapy, and antiangiogenic therapy) [106]. Lu et al. [10] studied the expression patterns of lactate metabolism-related genes in gliomas and their impact on molecular subtypes, patient prognosis, and immune therapy responses using single-cell and bulk RNA sequencing data. The results showed that lactate metabolism-related genes could be divided into A and B cluster subtypes in gliomas, with LDHA expressed in all six cell types and the lactate fraction being the highest in small glial cells. Patients in cluster B exhibited poorer survival rates and greater immune cell infiltration than those in cluster A, indicating the impact of lactate metabolism-related genes on glioma molecular subtypes and patient prognosis. A lactate metabolism score constructed through univariate Cox regression analysis and the PCA method could predict patient prognosis, with patients with higher lactate metabolism scores showing lower survival rates, providing a new approach for the personalized treatment of gliomas.

Several studies have revealed the importance of lactylation in GBM through in vivo and in vitro experiments. Lactate-mediated lactylation in tumor-associated macrophages (TAMs) enhances the secretion of immunosuppressive factors, leading to the reshaping of the tumor microenvironment (TME) and affecting the efficacy of immunotherapy. Early TAMs are mainly resident microglia (MGs), whereas later TAMs are replaced by monocyte-derived macrophages (MDMs). It has been found that MDMs disrupt glucose metabolism by activating the PERK-ATF4 pathway, driving immunosuppressive programs through histone lactylation, increasing IL10 expression, and suppressing T cell activity. Inhibition of lactate generation or lactylation can reduce IL10 expression, enhancing the efficacy of immunotherapy [69]. Tumor-derived lactate increases H3K18la levels, activating CD39 and CD73 gene transcription, which convert extracellular ATP into adenosine. The LDHA inhibitor, oxamate, reverses CD39 and CD73 expression, reducing lactate and adenosine generation. H3K18la also upregulates Tregs via the CCR8 pathway, leading to Th17/Treg imbalance and enhanced immune suppression. The combination of oxamate and chimeric antigen receptor (CAR)-T immunotherapy reduces Treg activation and enhances tumor-infiltrating lymphocyte (TIL) activity, indicating potential benefits in GBM treatment [102].

The self-renewal ability of glioma stem cells (GSCs) is closely related to tumor regeneration, treatment resistance, and heterogeneity maintenance, making them key targets for GBM treatment [107,108]. Histone lactylation, such as H3K18la, plays an important role in regulating GSCs self-renewal. NF-κB promotes lactate production by enhancing the Warburg effect, increasing the enrichment of H3K18la at the NF-κB-related LINC01127 promoter, thereby activating the transcription of these genes. LINC01127 overexpression activates the MAP4K4/JNK axis, mediating GSCs self-renewal and tumor progression. Targeting H3K18la can disrupt GSCs self-renewal, representing a potential approach for GBM treatment [101]. In addition, lactylation was associated with temozolomide (TMZ) resistance [70]. Studies have shown that lactate and H3K9la levels are elevated in recurrent and long-term GBM cells, conferring temozolomide (TMZ) resistance by activating *LUC7L2* transcription. The antiepileptic drug, stiripentol, significantly enhanced the sensitivity of GBM cells to TMZ by inhibiting lactylation, providing a new approach for improving GBM prognosis.

Lactylation also promotes vasculogenic mimicry (VM) formation, increasing tumor cell blood supply, and resulting in poor anti-angiogenic therapy efficacy [103]. Krüppel-like factor 15 phosphorylates and promotes LDHA transcription in the nucleus under the action of P4-135aa (a peptide encoded by MAPK6P4). Elevated LDHA expression mediates the lactylation of vascular endothelial-cadherin (cadherin) and vascular endothelial growth factor receptor 2 (VEGFR2), inducing VM in GBM. The knockout of these molecules can effectively inhibit VM and prolong survival in animal models. In summary, lactylation plays a key role in TME remodeling in GBM, GSC self-renewal, TMZ resistance, and VM formation, making it a novel target for GBM treatment (Table 2).

### 3.6. Lactylation and Hypoxia-Related Brain Damages

Recent studies have highlighted the critical role of lactylation in the regulation of neuronal apoptosis during cerebral ischemia/reperfusion injury (CIRI). CIRI involves the interruption and restoration of blood supply to the brain, leading to Ca^2+^ overload, inflammation, and mitochondrial dysfunction [109,110]. Yao et al. [13] reported elevated protein lactylation in the brain endothelium of rats with CIRI. Proteomic analyses identified 1003 lactylation sites in 469 proteins. KEGG and STRING analyses revealed that proteins such as VDAC1, SLC25A4, and SLC25A5, which are involved in Ca^2+^ signaling and mitochondrial function, were affected. SLC25A4 and SLC25A5 showed increased lactylation in CIRI rats, whereas VDAC1 lactylation was observed only in controls. These results suggest that lactylation contributes to CIRI via Ca^2+^ overload. In vivo experiments have shown reduced VDAC1 lactylation in CIRI rats, supporting the role of lactylation in mitochondrial and neuronal apoptosis.

Lymphocyte cytoplasmic protein 1 (LCP1) was upregulated in middle cerebral artery occlusion (MCAO) rats and oxygen-glucose deprivation/reperfusion (OGD/R)-stimulated PC12 cells. LCP1 knockdown reduces cerebral infarction (CI) injury. Lactylation increases LCP1 stability and promotes CI development. Both in vitro and in vivo models have shown increased LCP1 lactylation in CI. Inhibition of glycolysis via 2-DG reduced LCP1 lactylation and stability, indicating that lactate-mediated LCP1 lactylation facilitates CI progression by enhancing its stability. However, further studies are required to understand the role of LCP1 degradation in cerebral ischemia [104]. Acute ischemic stroke involves mitochondrial dysfunction, and astrocytes protect neurons by transferring functional mitochondria. Hypoxia/ischemia-induced lactate and lactylation reduce mitochondrial transfer in astrocytes. The lactylation of ADP-ribosylation factor 1 (ARF1) at K73 inhibits mitochondrial release. Low-density lipoprotein receptor-related protein 1 (LRP1) reduces lactate production and ARF1K73la expression, promoting mitochondrial transfer and neuroprotection in CIRI [105]. Further research on LRP1’s regulation of astrocyte metabolism could provide new therapies for CIRI (Table 2).

Current research also suggests that lactylation may represent a novel post-translational modification that regulates other hypoxia-related brain injuries, such as hypoxic-ischemic encephalopathy (HIE), global cerebral ischemia (GCI), and high-altitude brain edema. Lactylation may participate in the regulation of energy metabolism, oxidative stress, and inflammatory responses during hypoxic brain damage through mechanisms involving gene expression, protein function, and metabolic adaptation.

#### 3.6.1. Energy Metabolism Regulation

Hypoxia influences various physiological processes, particularly cellular energy metabolism. Cells can detect the oxygen supply status and adjust their metabolic pathways by enhancing glycolysis or limiting OXPHOS. This modulation helps in energy consumption, promoting metabolic adaptation, and mitigating energy crises. Lactylation predominantly targets glycolytic enzymes, inhibiting enzyme activity and reducing glycolytic product output [58]. Notably, increased H4K12la levels have been observed in astrocytes from AD models, with enrichment on the promoters of glycolytic genes such as HIF-1α, PKM, and LDHA. This enrichment promotes transcription of these genes, forming a glycolysis/H4K12la/PKM2 feedback loop, highlighting the role of lactylation in regulation [92].

Hypoxia-inducible factors (HIFs) are crucial for regulating the expression of genes related to glycolysis and lactate production, enabling cells to rapidly generate ATP under low-oxygen conditions and adapt to hypoxic environments [111]. Previous studies have established that HIF-1α stability is modulated by various PTMs, including acetylation, hydroxylation, ubiquitination, and phosphorylation [112,113]. Recent findings in prostate cancer research suggest that lactate can promote lactylation of HIF-1α, enhancing the transcription of *KIAA1199* and promoting angiogenesis [114]. This implies lactylation may coordinate with other PTMs to regulate HIF-1α stability [115]. However, the specific role and mechanism of HIF-1α lactylation in neural glycolysis demand further investigation.

In muscle cells, lactylation of mitochondrial proteins such as PDHA1 and CPT2 helps balance the reduced oxygen supply and oxidative phosphorylation activity, enabling cells to adapt to fluctuating oxygen levels and reduce energy consumption during high-intensity endurance activities [116]. Additionally, lactylation may enhance glycolysis by modulating glycolysis-related signaling pathways. H3K18la activates the AKT-mTOR signaling pathway by upregulating *VCAM1* transcription, promoting tumor cell proliferation, transformation, and migration. This highlights the role of the histone Kla/VCAM1/AKT-mTOR pathway in metabolic regulation [117]. Histone lactylation can also enhance USP39 expression, which interacts with the key glycolytic enzyme phosphoglycerate kinase 1 (PGK1) to promote its stability and deubiquitination, thereby increasing PGK1’s biological activity and potentially accelerating the glycolytic pathway. PGK1 is also implicated in the PI3K/AKT/HIF-1α signaling pathway, playing a crucial role in cell survival, proliferation, and metabolism [118]. These mechanisms collectively facilitate the transition of cellular metabolism under hypoxic conditions, aiding cellular adaptation to hypoxia.

#### 3.6.2. Oxidative Stress Management

Oxidative stress is a critical pathophysiological process that occurs during brain hypoxia-ischemia and subsequent reperfusion. In the early stages of hypoxia, oxidants produced by mitochondria and xanthine oxidase, along with excitatory amino acids, such as glutamine, lead to excitotoxicity and severe neuronal damage. During reperfusion, a substantial generation of oxidants, coupled with impaired antioxidant enzyme defense mechanisms, accelerates mitochondrial dysfunction and promotes cell apoptosis. Consequently, targeting oxidative stress is a pivotal therapeutic strategy for treating hypoxia-related brain disorders [119,120].

Under hypoxic conditions, cells reduce reactive oxygen species (ROS) levels and oxidative stress damage by inhibiting OXPHOS. Studies have demonstrated that the accumulation of alanyl-tRNA synthetase 2 (AARS2) mediates lactylation of mitochondrial proteins PDHA1 and CPT2 in mouse muscle cells, rendering these proteins inactive. This limitation of oxidative phosphorylation regulates the cellular oxygen balance. Furthermore, Aars2-/- mice exhibited elevated ROS levels in muscles during endurance running and increased levels of the ROS-induced oxidative damage marker malondialdehyde (MDA) upon exhaustion. These findings highlight the potential role of lactylation in the antioxidant stress response in hypoxia-related diseases [116].

#### 3.6.3. Inflammatory Response Modulation

Lactylation plays a crucial role in regulating inflammation-related gene expression, inhibiting excessive inflammatory responses, and promoting inflammation resolution, thereby mitigating neuronal damage. During the early stages of inflammation, external stimuli induce M1 macrophages to engage in aerobic glycolysis, leading to lactic acid accumulation, which promotes histone lactylation. In later stages, increased H3K18la regulates the transcription of repair genes such as *Arg1*, facilitating the transition of macrophages from the M1 to M2 phenotype and enhancing tissue repair [8]. In a rat model of myocardial ischemia, H3K18la levels peaked within 4 h post-myocardial ischemia reperfusion and subsequently declined. Concurrently, the mRNA levels of repair genes and H3K18la binding regions in monocytes were significantly upregulated. Monocyte reinfusion experiments have demonstrated that monocytes with elevated histone lactylation levels can reduce myocardial infarction size, decrease inflammatory cell infiltration, and improve cardiac function, underscoring the positive role of monocyte histone lactylation in immune homeostasis, tissue repair, and cardiac function during ischemia reperfusion [121]. Thibaut et al. [122] elucidated dynamic alterations in lactylation and gene expression during macrophage polarization in muscle tissue following ischemic injury. Their study indicated that dynamic changes in macrophage H3K18la marks at enhancers and promoters steered the transition from a proinflammatory to a prohealing phenotype. These investigations highlight the dynamic patterns of histone lactylation in monocytes and macrophages, providing a novel perspective on metabolic–epigenetic–immune cascade mechanisms, consistent with the findings of Zhang et al. [8].

However, lactylation also has deleterious effects. In renal proximal tubular cells (PTCs) subjected to ischemia-reperfusion, PFKFB3 expression is upregulated, promoting inflammation by mediating gene transcription through H4K12la and activating the NF-κB signaling pathway, thereby accelerating kidney fibrosis [123]. In macrophages with sepsis, HMGB1 lactylation enhances exosome secretion, promoting inflammation [124]. A similar mechanism has been observed in hepatocyte ischemia-reperfusion injury, in which the lactylation of HMGB1 promotes macrophage chemotaxis. Interventions such as blocking lactate production or using P300 inhibitors can reduce HMGB1 lactylation and subsequent secretion, thereby decreasing macrophage chemotaxis [125]. The dual role of lactylation may be attributed to its effects on different cell types and different stages of inflammation.

The human body functions as an integrated system, where the brain does not operate in isolation but interacts with multiple organs and systems, including the heart, lungs, kidneys, and immune system [126,127,128,129]. In some systemic injury factors, such as hypoxia-induced brain damage, the integrity of the blood–brain barrier (BBB) is compromised. Meanwhile, injury factors-enhanced peripheral lactylation will induce pro-inflammatory factors such as HMGB1 [124,125], or anti-inflammatory factors, as well as polarized macrophages [124], which can enter the central nervous system through damaged BBB, potentially exerting both positive and negative effects on brain injury. This hypothesis merits further exploration, and additional evidence is needed to validate its veracity.

## 4. Targeted Therapy

With the in-depth study of lactylation mechanisms and functions, targeting lactylation has emerged as a novel strategy for treating tumors, metabolic diseases, and inflammation-related conditions. The lactylation process involves lactate generation, shuttling, receptor binding, and regulation by “writers” and “erasers”, all of which represent potential therapeutic targets [115,130,131].

Targeting key glycolytic enzymes such as LDHA, HK, pyruvate dehydrogenase kinase (PDK), PKM2, and PFKFB3 to modulate lactate production has shown promise in cancer treatment, with several drugs advancing to clinical trials. LDHA inhibitors such as gossypol and its isomer AT-101 have shown significant effects in phase II clinical trials, both as monotherapy and in combination with standard chemoradiotherapy [132]. Stiripentol, an FDA-approved antiepileptic drug for Dravet syndrome, and its analogs (e.g., isosafrole), act as non-specific LDH inhibitors, blocking the conversion of pyruvate to lactate and exerting antiepileptic effects [133]. The glycolytic inhibitor 2-DG has demonstrated safety and efficiency in phase I/II trials (NCT00096707 and NCT00633087) for prostate cancer and advanced solid tumors [134,135]. In healthy brains with an intact BBB, 2-DG can cross the BBB via glucose transporters, suggesting its potential for treating central nervous system diseases. 2-DG reduces lactate accumulation and H3K9 lactylation, alleviating behavioral changes in MK801-induced schizophrenia (SCZ) mice [98]. Dichloroacetate (DCA) enhances mitochondrial glucose oxidation and reduces lactate production by inhibiting PDK. A phase I trial (NCT01029925) indicated that DCA is feasible and well tolerated in patients with recurrent glioblastoma and brain metastases [136]. A phase II trial (NCT01386632) demonstrated the safety of combining DCA with chemoradiotherapy for advanced head and neck squamous cell carcinoma, without adverse effects on survival and metabolites [137]. In AD research, disrupting the glycolysis/H4K12la/PKM2 positive feedback loop with PKM2 inhibitors, such as shikonin or compound 3 K, reduced lactate and H4K12la levels, inhibited pro-inflammatory microglial activation, and improved spatial learning and memory in AD mice [92]. Additionally, inhibiting PFKFB3 with the small-molecule inhibitor 3PO can suppress H4K12la and reduce the excessive transcription of NF-κB-related genes involved in chronic kidney disease fibrosis [123].

Targeting monocarboxylate transporters (MCTs) effectively blocks lactate uptake. AZD3965, a dual MCT1 and MCT2 inhibitor, showed positive therapeutic effects in late-stage solid tumors and B-cell lymphoma in a phase I trial (NCT01791595) [138]. The MCT1 and MCT4 molecular chaperone inhibitors, CD147 and meplazumab, demonstrated safety and tolerability in phase I and II trials in healthy volunteers and COVID-19 patients [139]. Additionally, two clinical trials on the GPR81 inhibitor curcumin are ongoing: one assessing its safety in children with acute lymphoblastic leukemia during chemotherapy maintenance (NCT05045443) and another exploring whether curcumin combined with piperine can delay or prevent progression in prostate cancer, monoclonal gammopathy of undetermined significance, or specific myelomas (NCT04731844). The MCT1 inhibitor CHC can alleviate lactate-induced Snail1 lactylation and activation of TGF-β/Smad2 after hypoxia, reducing endothelial-to-mesenchymal transition, myocardial fibrosis, and cardiac dysfunction post-myocardial infarction [67].

Furthermore, “writers” and “erasers” directly influence lactylation formation and removal, presenting new therapeutic opportunities. The CBP/p300 inhibitor, C646, reduces inflammation in hepatic ischemia-reperfusion injury by inhibiting HMGB1 lactylation [125], whereas A-485 exhibits anti-angiogenic effects on proliferative retinopathy by reducing YY1 lactylation [140]. The HDAC1-3 inhibitor MS-275 (entinostat) significantly increased H3K14la and H3K18la levels during early development [71]. In mouse hematopoietic stem cells, HDAC inhibitors such as apicidin and MS-275 increased H3K18ac but decreased H3K18la levels due to acetylation crosstalk, reducing fibrosis-related gene expression [141]. HDAC inhibitors, such as vorinostat, romidepsin, belinostat, and tucidinostat, have been approved for the treatment of lymphomas [142]. SIRT3, which delactylates CCNE2, may represent a new target for liver cancer treatment [143]. Targeted lactylation has significant potential for disease treatment. Ongoing research and clinical trials of various drugs and targets have made substantial progress. However, lactylation shares “writers” and “erasers” with other post-translational modifications, particularly acetylation, raising the risk of off-target effects. As the molecular and regulatory mechanisms of lactylation in diseases become clearer, the development of more precise targeted drugs is anticipated.

## 5. Detection of Protein Lactylation

Zhang et al. [8] employed HPLC-MS/MS to identify 26 proteins and 16 lactylation sites on histones in humans and mice, respectively. This methodological approach, involving protein extraction, enzyme digestion, modified peptide enrichment, and liquid chromatography–mass spectrometry analysis, provides a robust foundation for lactylation research. This technology facilitates the mapping of lactylated proteins across various biological systems, including human lung tissues, human gastric cancer cells, animal models, plants, parasites, and bacteria [13,59,60,61,62,63,144]. The expansion of lactylation datasets has enabled machine learning to predict new lactylation sites by identifying characteristic patterns, offering a more efficient alternative to traditional experimental detection, particularly for large-scale screening (Figure 3).

Jiang et al. [145] developed a novel predictor, FSL-Kla, based on a few-shot learning architecture, using 343 lactylation sites from 191 proteins across humans, mice, and Botryotinia fuckeliana as reference data. The model, trained and optimized using the KSP, integrates seven sequence-based and three structure-based features. This multi-feature hybrid ensemble system significantly enhanced prediction performance, achieving a 16.2% increase in the area under the ROC curve (AUC) for the in vivo prediction of lactylation sites. Despite its promise, FSL-Kla currently identifies only a limited number of lactylation sites and may lack accuracy in complex scenarios, necessitating further experimental validation (Figure 3).

Deep-Kla was the first computational model for identifying lactylation sites in rice proteins [146]. Using 172 rice protein lactylation sites as a benchmark and 273 lactylation sites from *Botrytis cinerea* as test data, the model leveraged amino acid sequences around lysine K. Positive and negative samples were balanced, and redundant sequences were removed using the CD-HIT program. Deep Kla’s architecture comprises four interconnected subnetworks: a word embedding layer, a convolutional neural network (CNN), a bidirectional gated recurrent unit (BiGRU), and an attention mechanism layer. This design allows for the automatic extraction of sequence features and capture of long-range and key positional information within protein sequences. Deep-Kla demonstrates powerful predictive capabilities and is accessible via its web server (http://lin-group.cn/server/DeepKla, accessed on 10 July 2024) (Figure 3). Both FSL-Kla and Deep-Kla have marked a significant transition from basic experimental research to computational predictions of lactylation sites, enhancing efficiency and reducing experimental costs. However, the limited dataset sizes and early stages of lactylation research pose challenges, including potential underfitting or overfitting, owing to insufficient training data. Machine learning algorithm performance is highly dependent on design choices, such as neural network architectures, training procedures, regularization methods, and hyperparameters.

Lai and Gao [147] addressed these challenges using Auto-Kla, a novel automated machine learning server. Auto-Kla incorporates a larger sample size than FSL-Kla and Deep Kla and leverages advanced models and algorithms to automatically identify lactylation sites without manual sequence input. Using a reference dataset of 1014 protein sequences from gastric cancer cells and 2375 lactylation sites, Auto-Kla employed stratified sampling and transformers with CNN-BiGRU attention mechanisms. Training and evaluation were conducted using 10-fold cross-validation, demonstrating superior performance and reliability compared to existing models. Auto-Kla also exhibited strong performance on other PTM datasets, confirming its universality and portability. See https://github.com/tubic/Auto-Kla (accessed on 10 July 2024) for details of the web server (Figure 3).

## 6. Conclusions and Perspectives

Lactylation, an emerging post-translational modification, offers promising avenues for investigating both physiological functions and pathological conditions in the human body, with significant clinical implications. Elucidating the role of lactylation as a linkage between NCC metabolic states, gene regulatory networks (GRN), and developmental gene expression expands our insights into neurocristopathies, such as orofacial clefts and congenital heart disease, potentially paving the way for novel metabolism-based strategies in developmental and regenerative medicine. Moreover, the identification of lactylation in CNS diseases underscores its potential as a biomarker for early diagnosis, treatment, and prognosis.

Moving forward, elucidating the precise molecular mechanisms of lactylation remains imperative. Current research indicates lactyl-CoA and LGSH as substrates for lactylation, yet exploring additional substrates could deepen our understanding of lactylation’s role in various diseases and physiological states. Regulatory factors governing lactylation likely involve numerous “writers” and “erasers”. For instance, HBO1 from the MYST family mediates H3K9la, influencing the transcription of tumor-associated genes [148], highlighting its role as a novel histone lactylation “writer”. Recent studies have also identified a p300-dependent interaction with the histone chaperone ASF1A in regulating H3K18la, impacting vascular endothelial injury [149]. Additionally, alanyl-tRNA synthetase (AARS1) acts as a lactate sensor and a transferase in global lysine lactylation in tumor cells [150], paralleled by AARS2 in mitochondrial protein lactylation [116]. In prokaryotes, YiaC and CobB serve as “writers” and “erasers” for protein lactylation [151], expanding our understanding of lactylation regulators and suggesting new therapeutic avenues.

Considering the complexity of epigenetic modifications, interventions targeting lactylation may inadvertently affect other acylation modifications, leading to off-target effects. Therefore, further safety studies are crucial to confirm the viability of these strategies in clinical applications. This review investigates the significance of lactylation modification in lactate metabolism, its underlying mechanisms, and its roles in neurodevelopment, neuropsychiatric disorders, nervous system tumors, and cerebrovascular diseases, thereby demonstrating the diverse effects of lactylation on brain function and disease progression. By elucidating the precise molecular mechanisms of lactylation and its specific effects on various stages of CNS diseases and cell types, we lay a robust foundation for precise neurological disease treatments through lactylation targeting.

## Figures and Tables

**Figure 1 biomolecules-14-01175-f001:**
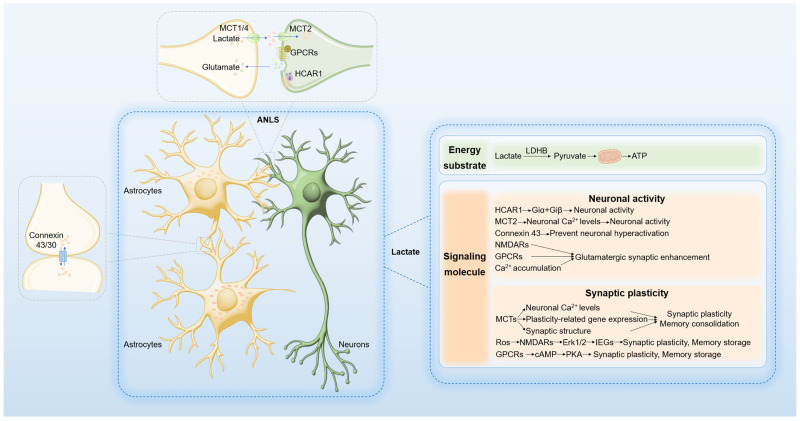
Lactate shuttling and its functional mechanism. Following neuronal activation, glycolysis is enhanced, leading to an increased release of glutamine from activated synapses, which is subsequently taken up by astrocytes. This process stimulates both glucose uptake and lactate production in astrocytes. Lactate produced by astrocytes is shuttled between neurons and astrocytes, as well as between astrocytes themselves, via monocarboxylate transporters (MCTs) and gap junction hemichannels (such as Connexin 43 and Connexin 30). Lactate serves not only as an energy substrate but also as a signaling molecule that modulates neuronal activity and synaptic plasticity.

**Figure 2 biomolecules-14-01175-f002:**
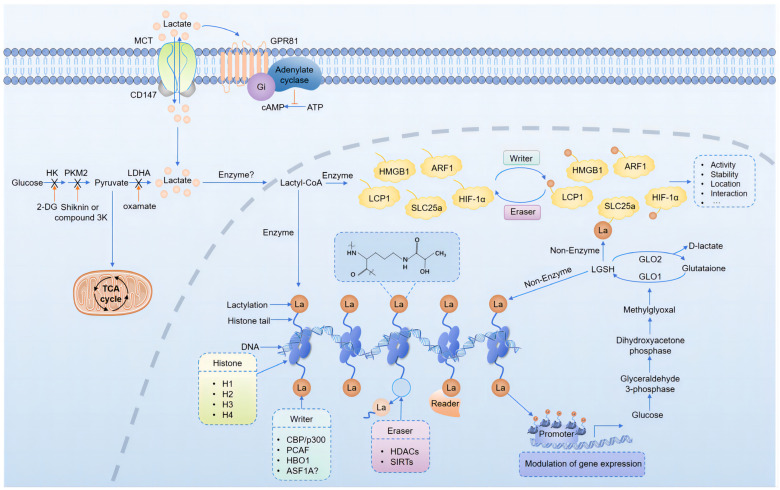
Mechanisms and regulation of lactylation. Lactate shuttles between cells via MCTs and enters cells as an energy substrate. It also acts as a signaling molecule by binding to the cell surface receptor GPR81, mediating cell communication. Once activated to lactyl-CoA, lactate serves as a donor for lactyl groups. “Writers” and “erasers” add or remove these lactyl groups from amino acid residues of histone and non-histone proteins, forming lactylation modifications, while “readers” interpret this epigenetic information. This process influences gene transcription, as well as protein activation, stability, localization, and interactions. Gaffney et al. [58] proposed a mechanism where LGSH acts as a substrate in non-enzymatic protein lactylation. Various glycolytic enzymes involved in lactate production also regulate lactylation.

**Figure 3 biomolecules-14-01175-f003:**
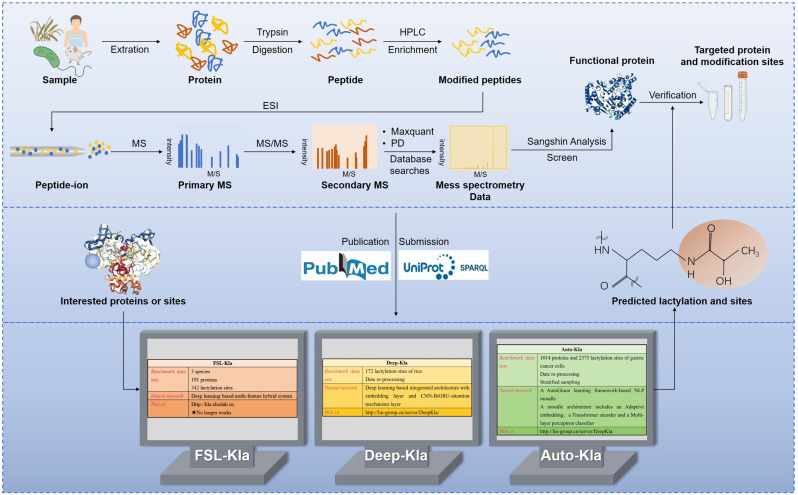
Workflow for the detection of lactylation. Proteins are extracted from tissues or cells and digested into peptides using Trypsin. HPLC pre-fractionates the peptides, enriching the modified ones. ESI or MALDI ionizes the peptides, forming positively charged ions, which are analyzed by their mass-to-charge ratio (*m*/*z*) in the mass spectrometer, producing a primary mass spectrum. The most intense precursor ions are selected for fragmentation to generate a secondary mass spectrum. Bioinformatics analysis identifies functional proteins and modification sites. Validation is performed using Western blot, IHC, and ELISA. Utilizing data on lactylated proteins and sites, machine learning models (e.g., FSL-Kla, Deep-Kla, Auto-Kla) are developed to identify and predict lactylation sites. (“*” The website is no longer operational).

**Table 1 biomolecules-14-01175-t001:** Lactylation in the physiological processes of central nervous system.

Physiological Process	Lactylation Proteins	Protein Targets	Regulation	Key Findings	References
Neural development	H3K18	Genes involved in neuronal differentiation and maturation	HDAC1-3 inhibition Induces genome-wide enhancement of H3K18la	Unraveling the dynamics of histones Kcr and Kla and their functions in neural development	[71]
	H3K18	Pluripotency genes and “second wave” genes	Glis1 upregulates the expression of glycolysis-related genes	The central role of histone Kac and Kla in epigenetic regulation driven by glycolytic metabolism	[78]
	H3K18	Genes that contribute to NCC-specific features	Genomic regions containing *SOX* and *TEAD* motifs tend to be lactylated in NCC	The role of lactylation as a mechanism linking NCC metabolic states with GRN and developmental gene expression	[11]
	Histone	LRRC15	Bach1 suppresses HK2 and GAPDH expression, thereby downregulating glycolysis	Maintenance of microglia metabolic homeostasis is important to astrocytogenesis during early brain development	[80]

**Table 2 biomolecules-14-01175-t002:** Lactylation in central nervous system diseases.

		Lactylation Proteins	Protein Targets	Regulation	Key Findings	References
Neuropsychiatric disorders	Depression and Anxiety	H1	C-Fos	4-CIN, Oxamate	Stress-associated neural excitation stimulates H1 lactylation, which correlates with decreased social behavior.	[81]
	AD	H4K12	PKM2	Glycolysis/H4K12la/PKM2 positive feedback loop	Glycolysis/H4K12la/PKM2 positive feedback loop exacerbates microglial dysfunction.	[92]
		H3K18	Rela NF-κB	Lactate	H3K18la/NF-κB/SASP positive feedback loop exacerbates AD pathology.	[68]
		H3	Arg1 VEGF	Exercise-induced Lactate	Exercise-induced lactate directs microglia to a repair/anti-inflammatory state via H3Lactylation	[96]
	PD	H3K9	Slc7a11	2-DG	H3K9la enhances transcription of *Slc7a11*, promoting pro-inflammatory microglial activation	[97]
	SCZ	H3K9 H3K18	Mybe HMGB1	2-DG	In SCZ model, both glycolysis and lactylation were elevated, these increases could be inhibited by 2-DG	[98]
	PTSD	HIF-1α			HIF-1α lactylation and dysfunction play a critical role in the pathological development of PTSD	[99]
Cerebral neoplasms	GBM	Histon	IL-10	The PERK-ATF4-driven glucose metabolism	Histone lactylation drives immunosuppressive programs, increases IL-10 expression, and suppresses T cell activity	[69]
		H3K18	LINC01127	NF-κB promotes lactate production	H3K18la plays an important role in regulating GSCs self-renewal.	[101]
		H3K18	CD39, CD73	Oxamate	H3K18la enhances immune suppression	[102]
		H3K9	LUC7L2		Stiripentol enhances the sensitivity of GBM cells to TMZ by inhibiting H3 lactylation	[70]
		VE-cadherin VEGFR2		P4-135aa mediates KLF15 phosphorylation, promoting LDHA transcription and facilitating VE-cadherin and VEGFR2 lactylation.	The lactylation of VE-cadherin and VEGFR2 induces VM formation in GBM	[103]
Hypoxia-related brain damages	Stroke (CIRI)	496 proteins 1003 sites			Elevated lactylation in CIRI rats contributes to mitochondrial dysfunction and neuronal apoptosis through Ca^2+^ overload	[13]
		LCP1		2-DG reduces LCP1 lactylation and stability	LCP1 lactylation facilitates CI progression by enhancing its stability	[104]
		ARF1K73		LRP1 reduces ARF1- K73la	LRP1 reduces ARF1K73la, promoting mitochondrial transfer and neuroprotection in CIRI	[105]

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
