# Peer review of "Lactylation: A Novel Post-Translational Modification with Clinical Implications in CNS Diseases"

_biomolecules, 2024, doi:10.3390/biom14091175_

Round 1
Reviewer 1 Report
Comments and Suggestions for Authors
Manuscript ID.: biomolecules-3131837
Title: Lactylation: A Novel Post-Translational Modification with Clinical Implications in CNS Diseases Clinical Implications in CNS Diseases.
The review provides valuable insights into post-translational modifications (PTMs) and highlights the promising potential of lactylation in central nervous system (CNS) diseases. However, the extensive and complex information presented may overwhelm readers and make the paper seem inaccessible. To enhance clarity and engagement, the author should focus on the key points related to the review’s title. If the author is interested in discussing the mechanisms behind acetylation, it would be better to cover this in a separate review article. Additionally, increasing the use of diagrams and tables to present information could make the article more engaging and easier for readers to follow.
Comments on the Quality of English LanguageYes
Author Response
Dear reviewer,
Thank you for reviewing our manuscript and for the constructive comments, which greatly helped us to improve the manuscript. The manuscript was carefully revised and point-by-point response was listed below. The major revision has been highlighted in red.
Comments 1: The review provides valuable insights into post-translational modifications (PTMs) and highlights the promising potential of lactylation in central nervous system (CNS) diseases. However, the extensive and complex information presented may overwhelm readers and make the paper seem inaccessible.
Response 1: Thank you very much for your valuable suggestion. To address your concern regarding the extensive and complex information presented in the manuscript, we have restructured the paper. We merged the original Section 2, "Lactate Metabolism in the Brain" and Section 3, "Lactate as a Signaling Molecule in the Brain" into a single section titled "Lactate Metabolism and Shuttling in the Brain" (revised Section 2) (P2L58, in red). We have also added subsections 2.1 "Lactate as an energy substrate in the brain" and 2.2 "Lactate as a signaling molecule in the brain" to enhance the clarity (P2L73, P3L129, in red). Additionally, we included a diagram (P5, Figure.1, in red) to visually represent the content of Section 2. To provide readers with a more coherent understanding of lactylation in biological processes, we have moved the original Section 4.3, "Detections of protein lactylation" to the end of the manuscript (revised Section 5) (P18L740-P19L797, in red), discussing it as a separate topic. Corresponding adjustments have been made to other sections to ensure a smooth flow throughout the manuscript. We hope these revisions will enhance the clarity and accessibility of the content.
Comments 2: To enhance clarity and engagement, the author should focus on the key points related to the review’s title. If the author is interested in discussing the mechanisms behind acetylation, it would be better to cover this in a separate review article.
Response 2: Thank you very much for your valuable suggestion. We have removed the content related to the mechanisms of acetylation to maintain a clear focus on the key points of the present manuscript (P8L339-P8L347, in red).
Comments 3: Additionally, increasing the use of diagrams and tables to present information could make the article more engaging and easier for readers to follow.
Response 3: Thank you very much for your suggestion. In response, we have summarized and illustrated the content of Section 2, "Lactate metabolism and shuttling in the Brain" with a diagram (P5, Figure.1, in red). Now the article includes 3 figures and 2 tables, which are helpful to improve the clarity and accessibility of the manuscript.
Thank you again for your thorough and insightful review. We hope that the revised manuscript meets your satisfaction and look forward to hearing back from you.
Sincerely yours,
Yi Qu
Yi Qu, PhD
Professor of Department of Pediatrics, West China Second University Hospital
Sichuan University, Chengdu, Sichuan 610041, China
Fax: +86-28-85559065
Email: quyi@scu.edu.cn
Reviewer 2 Report
Comments and Suggestions for Authors Lactylation is a novel topic, this work clearly explains the role of this procedure in the central nervous system. I think that something could be mentioned about lactylation at the peripheral level that has an impact on the central nervous system, to clarify the importance of this process. It might be more illustrative to place a figure in the section on lactate as a signaling molecule. To clarify the impact on diseases due to metabolic dysregulation. It is important to be careful when mentioning abbreviations, since for example the abbreviations DAB and HCAR are not mentioned but are widely used, and some abbreviations are not clarified. The rearrangement of some sections of the work seems pertinent, such as the detection of protein lactylation, which I consider could be left at the end of all the sections as a separate topic since the continuity of the role of lactylation in biological processes is lost. The central theme of the review is lactate, but the mention of its relationship with other metabolic pathways may be relevant for the interconnection of metabolic pathways where lactate has an impact in addition to the post-translational modification it has.Author Response
Dear reviewer,
Thank you for reviewing our manuscript and for the constructive comments, which greatly helped us to improve the manuscript. The manuscript was carefully revised and point-by-point response was listed below. The major revision has been highlighted in red.
Comment 1: I think that something could be mentioned about lactylation at the peripheral level that has an impact on the central nervous system, to clarify the importance of this process.
Response 1: Thank you very much for your valuable suggestion. We appreciate the importance of discussing lactylation at the peripheral level and its impact on the central nervous system. In Section 4.7.3 of the original manuscript, we addressed peripheral lactylation. Based on your constructive feedback, we have revised this section to emphasize the connection between peripheral lactylation and its overall significance in the context of central nervous system function in the manuscript (P16L670-P17L674, P17L675-P17L678, in red).
Comment 2:It might be more illustrative to place a figure in the section on lactate as a signaling molecule. To clarify the impact on diseases due to metabolic dysregulation.
Response 2: Thank you very much for your valuable suggestion. We have added a figure to illustrate lactate as a signaling molecule in the revised manuscript (P5, Figure.1, in red), focusing on lactate shuttling and its functional mechanism.
Comment 3: It is important to be careful when mentioning abbreviations, since for example the abbreviations DAB and HCAR are not mentioned but are widely used, and some abbreviations are not clarified.
Response 3: Thank you for your careful check. We are apologized for our negligence. We have provided the full name of the abbreviations, including DAB (P4L187, in red) and HCAR (P3L143, in red).
Comment 4: The rearrangement of some sections of the work seems pertinent, such as the detection of protein lactylation, which I consider could be left at the end of all the sections as a separate topic since the continuity of the role of lactylation in biological processes is lost.
Response 4: Thank you very much for your suggestion. We have moved the section on the detection of protein lactylation to the end of the manuscript to maintain the continuity of the role of lactylation in biological processes in the revised manuscript (revised Section 5) (P18L740-P19L797, in red).
Comment 5:The central theme of the review is lactate, but the mention of its relationship with other metabolic pathways may be relevant for the interconnection of metabolic pathways where lactate has an impact in addition to the post-translational modification it has.
Response 5: Thank you for your professional review work. We greatly appreciate your insights regarding the relationship between lactate and other metabolic pathways, as well as its significance in understanding the broader metabolic network. However, considering that the central theme of our review is lactylation, our primary goal is to focus on this specific post-translational modification and its roles and mechanisms in biological processes. We believe that adding a discussion on the relationship between lactate and other metabolic pathways may divert the focus of the manuscript from the core topic we aim to convey. That said, we have ensured that the importance and background of lactylation are clearly articulated in the text, so readers can fully appreciate its unique role.
Thank you again for your thorough and insightful review. We hope that the correction will meet with approval.
Sincerely yours,
Yi Qu
Yi Qu, PhD
Professor of Department of Pediatrics, West China Second University Hospital
Sichuan University, Chengdu, Sichuan 610041, China
Fax: +86-28-85559065
Email: quyi@scu.edu.cn
Reviewer 3 Report
Comments and Suggestions for Authors
Very comprehensive review of what appears to be a timely and extensive compilation of the current literature on lactate as marker, energy source, transcriptional regulator, epigenetic modifier, and stress/hypoxia/GBM modulator. I am not an expert on lactylation, so this was an interesting and well written review. Only a few minor comments: there a fair number of definitive statements such as line 201 ' molecular mechanisms reveal', and at the same time many somewhat vague statements regarding putative causal associations. The authors could review if the evidence is strong enough for definitive statements, and if it is more speculative then indicate it as so. A very few typos such as 264 functionality instead of functional, but in general English is excellent.
Comments on the Quality of English LanguageExcellent. Were any AI or chat bot language programs used in the manuscript, and if so should this be reported?
Author Response
Dear reviewer,
Thank you very much for taking your time to review this manuscript. We appreciate your thoughtful feedback and for your kind words about our manuscript. The manuscript was carefully revised and point-by-point response was listed below. The major revision has been highlighted in red.
Comment 1:There a fair number of definitive statements such as line 201 ' molecular mechanisms reveal', and at the same time many somewhat vague statements regarding putative causal associations. The authors could review if the evidence is strong enough for definitive statements, and if it is more speculative then indicate it as so.
Response 1:Thank you very much for your insightful comments. In response to your concerns, I have referenced additional studies from high-impact journals in the revised manuscript (P5L205-P5L209, in red). Extensive research has confirmed astrocyte-neuron lactate transport is essential for synaptic plasticity. The expression of pCREB and Arc, along with phosphorylated cofilin, represents key mechanisms underlying long-term synaptic plasticity and memory formation, as well as the associated synaptic structural changes, all of which hinge on lactate transport between astrocytes and neurons [Cell. 2011 Mar 4;144(5):810-23]. Further experiments demonstrate that lactate metabolism elevates the intracellular NADH/NAD ratio, enhancing neuronal redox state and potentiating NMDAR activation and subsequent Erk1/2 signaling through redox-sensitive NR1 subunits, promoting IEG expression. Specific inhibitors MK801 and U0126 abolish lactate-induced IEG expression [Proc Natl Acad Sci U S A. 2014 Aug 19;111(33):12228-33] [ Sci Rep. 2022 Nov 10;12(1):19238]. These studies provide substantial support for our assertion, demonstrating the molecular mechanisms through which lactate influences synaptic plasticity.
To enhance the quality of the manuscript, we have carefully reviewed the text to ensure that our claims are adequately supported by strong evidence. Additionally, we have examined sections where the evidence may have been perceived as insufficient or speculative. In these instances, we have incorporated qualifiers such as "may," "suggest," and "require further validation" to emphasize the tentative nature of these claims in the revised manuscript (P4L168, P4L174, P5L219, in red)
Comment 2:A very few typos such as 264 functionality instead of functional. but in general English is excellent.
Response 2:We were really sorry for our careless mistakes. Thank you for your reminder. We have carefully checked the manuscript and corrected the errors (P7L283, in red).
Comment 3: Were any AI or chat bot language programs used in the manuscript, and if so should this be reported?
Response 3: I would like to clarify that we did not utilize any AI or chat bot language programs in the preparation of the manuscript. However, we did engage Editage for professional language editing services, and the editing certification is provided as the attachment.
Thank you again for your thorough and insightful review. We hope that the correction will meet with approval..
Sincerely yours,
Yi Qu
Yi Qu, PhD
Professor of Department of Pediatrics, West China Second University Hospital
Sichuan University, Chengdu, Sichuan 610041, China
Fax: +86-28-85559065
Email: quyi@scu.edu.cn

Round 2
Reviewer 1 Report
Comments and Suggestions for Authors
No comment